# A Conceptual Framework of the Sustainability Challenges Experienced during the Life Cycle of Biobased Packaging Products

**Deniz Turkcu ***[iD]**, Nina Tura and Ville Ojanen**

Department of Industrial Engineering and Management, LUT University, 53851 Lappeenranta, Finland
* Correspondence: deniz.turkcu@lut.fi

**Abstract:** Biobased packaging products are framed as products that have environmental value. They are promoted by many institutions and companies as a way of addressing climate change challenges by decreasing carbon footprints and providing alternatives for the fossil fuel-based economy. The use of biobased packaging products has started to become widespread, and they are increasingly commercially available. Despite the acknowledged benefits of such products, there are several challenges associated with the use of them. This paper provides a state-of-the-art review of biobased packaging products and presents a conceptual framework of the sustainability challenges experienced over their life cycles. The framework categorizes the identified challenges by their environmental, social, and economic impact, as well in terms of the different life cycle stages, from beginning of life to middle of life to end of life. In addition to increasing the understanding of the challenges associated with biobased packaging materials and their use, the proposed framework benefits the analysis of these challenges in different organizations, the identification of potential greenwashing, and the development of mitigation strategies to overcome the challenges. Furthermore, this study reveals gaps in the literature to be considered in future research into biobased packaging products.

**Keywords:** biobased; packaging; biodegradable; sustainability; challenges; life cycle

## 1. Introduction

Consumer lifestyles have been changed by increasing urbanization, population and economic growth, and the rising standard of living. Moreover, the demand for pre-packaged, convenient, and single-serving foodstuffs, consumer goods, and healthcare/hygiene products, as well as the rapid growth in e-commerce [1], the aging population, and long and complex supply chains [2] are all increasing the demand for packaging. According to industry analyst Smithers Packaging (2019), the global packaging market increased by 8.4% to USD 914.7 billion from 2015 to 2019 and is expected to increase by 2.8% annually to USD 1.05 trillion in 2024 [3]. COVID-19 has specifically increased packaging consumption, as it has created an increasing trend toward internet shopping and food delivery. This trend is expected to continue in the coming years. Currently, packaging is a part of everyday life. This is especially true of plastic packaging, with consumption of this product type having increased significantly [4].

At present, the packaging industry heavily relies on petroleum-based plastics [5], which have many advantages in terms of being lightweight, durable, flexible, affordable, strong, and convenient [6]. However, plastic packaging is also associated with many disadvantages and challenges related to economic, environmental, social, and technical issues [7]. The main of these disadvantages stems from the possible scarcity of fossil fuels in the future and the production processes that contribute to increased greenhouse gas emissions in the value chain. New legislations, such as the banning of single-use plastic packaging, is also creating demand for alternative types of packaging. Indeed, societal demand for sustainable production and consumption is also growing due to

increasing awareness of the problems related to oil-based plastics, such as problematic waste management, the accumulation of plastics in our seas and oceans, and the negative health impacts on humans and animals. Consequently, the amendments being made to regulations alongside the increase in public pressure are creating a demand for the development of alternatives to fossil-based plastic packaging [6].

Renewable and biobased options are being developed for packaging as a solution to the challenges associated with conventional plastics. These alternatives are claimed to have advantages compared to fossil-based options, such as smaller carbon footprint, increased acceptance by consumers, and more sustainable end-of-life (EOL) options [4]. Moreover, it is claimed that biobased products will decrease the dependency on fossil fuels and the landfill of plastic solid waste [8]. Moreover, the price of biobased products depends on biomass, which a has more stable price compared to the fluctuations in oil prices [9]. The production costs associated with biobased products are also expected to decrease in the near future, as the demand for biobased products is expected to increase, allowing economies of scale [6].

Despite the multiple benefits, there are multiple challenges associated with biobased packaging products [6,8,10]. As these products are highly promoted as a part of the bioeconomy, sustainability-related risks and challenges should be considered in a holistic way [6]. These include challenges such as possible land use change from food production to biomass production [6,8,10–19], safety and health problems in production and processing [6,8,17], issues such as toxic pesticides [6,8,11,13,15–17], and the use of genetically modified organisms (GMO) in biomass production [6,8,17,19], and challenges related to the EOL of the products, such as consumers not knowing how to separate waste [6,17,20,21], and a lack of infrastructure [21]. The current literature acknowledges that these challenges should be considered throughout the products' value chains and from the various sustainability aspects, including environmental, economic, and social concerns [6,10]. However, there is a lack of research reviewing all three of these sustainability aspects in a holistic way, especially in relation to studies with a social perspective.

This study goes beyond the state of the art in this field (e.g., Gerassimidou et al. (2021), Spierling et al. (2018) [6,10]) by explaining how different challenges can interact within different sustainability domains over different life cycle phases. In this study, the sustainability challenges of biobased packaging products are considered in a holistic way, not only considering them in comparison to conventional plastics. This is because biobased plastics may also be used as an alternative to paper, glass, recycled plastic, or some other material. By analyzing these problems based on how they contribute to different sustainability domains within different life cycle stages, this study also aims to identify the most crucial issues that may work as barriers to the realization of the more demanding sustainable development objectives.

Two research questions were formed to address these research gaps:

1. What are the main environmental, social, and economic challenges associated with biobased packaging products in their different life cycle stages?
2. What are the missing parts and emerging themes in the current literature on the sustainability of biobased packaging products?

Based on the review of the current literature, this study introduces a conceptual framework of the challenges over the life cycle of biobased packaging products. This framework analyzes and categorizes the challenges in accordance with their sustainability aspects, using the triple bottom line concept through three dimensions of: the environmental, economic, and social [22], which are commonly used to analyze the sustainability of the companies and products [23,24]. Environmental sustainability is related to the efficiency of energy use, waste management and reduction, pollution reduction, and the use of hazardous/harmful/toxic materials [23]. Social sustainability relates to how people will be affected in terms of equality, health, diversity, connectedness, quality of life, democracy, and accountable governance [25]. Economic sustainability refers to balanced long-term growth without negative impacts on the environment, society, or company culture [26]. This study

contributes to the existing research by highlighting that different challenges of biobased packaging products can interact with each other across different sustainability dimensions, which emphasizes the need to create a holistic picture of the possible challenges. The prior literature focuses mainly on the challenges from economic and environmental perspectives, but we also highlight the challenges associated with the social dimension. By discussing the challenges associated with biobased packaging materials from multiple viewpoints, this study aims to help companies using such products identify these challenges and avoid possible greenwashing activities. Furthermore, by dividing the challenges according to the life cycle stages of the products as beginning of life (BOL), middle of life (MOL), and end of life (EOL) [27], we provide a tool for companies to develop mitigation and prevention measures targeted for actions in certain life cycle stages.

## 2. Background Literature

### 2.1. Defining Biobased Packaging

Currently, the largest field of use for biobased plastics is the packaging industry [17]. Biobased packaging products are products made from corn, sugar beets, bamboo, wood, and sugarcane. The most common biobased packaging products are biobased plastic packaging products made from sugarcane [6]. Biobased plastic packaging products are plastic products made of biological resources that are (partly) derived from renewable resources or the side streams of agri-food products [28]. The biomass type used in bioplastics is often sugarcane, cellulose, lignin, and corn, and they are mainly produced in Asia (followed by Europe, North America, and South Africa) [10]. The term 'bioplastic' or 'biobased plastic' is often used loosely and synonymously alongside 'biodegradable plastic'. However, even though some bioplastics are indeed biodegradable, not all are.

Biobased materials can be made from the first, second, or third generation of feedstocks. First-generation feedstocks are feedstocks such as corn, whey, and sugarcane [6]. Second-generation feedstocks are byproducts from the agricultural and forestry industries and municipal solid waste. Third-generation feedstocks include biomass from algae [29].

There exist different categories of biobased plastics:

1. Biobased/renewable and non-biodegradable plastic (such as polyethylene (PE), polypropylene (PP), or polyethylene terephthalate (PET)); and
2. Biobased/renewable and biodegradable plastic (such as polylactic acid (PLA), poly-hydroxyalkanoates (PHA), or polybutylene succinate (PBS)).

Biodegradable and compostable plastics can be broken down by microorganisms such as bacteria called Pseudomonas and fungal species such as Rhizopus delemar, Rhizophus arrizus, Aspergillus flavus, and Penicillium [30]. With the help of these organisms, plastics are broken into water, carbon dioxide, mineral salts, and new biomass within a defined period and under certain conditions. For a product to be called biodegradable or compostable, the material must comply with official standards of biodegradability, and there are certifications that prove the compostability or biodegradability of the products [31]. The temperature, duration, presence of microorganisms, nutrients, PH, oxygen, and moisture levels all affect if and how fast the material will biodegrade [32]. When a product composts, in addition to biodegradation, it also becomes available to add nutrition to the soil [33]. Hence, a material that degrades in industrial composting does not necessarily degrade in home composting, as different materials need specific conditions to biodegrade.

The biodegradability of biobased plastics is dependent on the chemical and physical structure of the monomer rather than the source material [34]. Surface conditions such as surface area, hydrophilic and hydrophobic properties, first-order structures such as the chemical structure, molecular weight, and molecular weight distribution, and high-order structures such as glass transition temperature, melting temperature, modules of elasticity, crystallinity, and crystal structure of the polymers play roles in determining if the polymers are biodegradable or not [35].

This paper only focuses on biobased plastics, both biodegradable and non-biodegradable. Even though there are petroleum-based plastics that can be biodegradable, they are not in the scope of this paper.

Many biobased materials are not made of 100% biobased products. They are often a mix of biobased and non-biobased materials. There are national regulations stating the required threshold for the renewable content of a product to be called biobased. However, there is no universal consensus on this matter [6]. There are labels that indicate the percentage of biomass content in the biobased material provided by certifiers such as DIN CERTCO and TUV AUSTRIA Belgium [36].

### 2.2. Sustainability and Life Cycle Management of Biobased Packaging

The biobased product industry is developing very quickly in terms of new materials and innovations. However, much of the research is not yet public, and there are still many products that are being developed and waiting for food contact approval, for example [4]. As the development of and challenges associated with these products are still an emerging research area, their impact is not yet well known. The environmental sustainability performance of biobased plastics is especially being scrutinized by different stakeholder groups, and the amount of literature on environmental concerns has been increasing [6,10], while studies considering the economic and social aspects remain very limited. So far, no fully biobased biodegradable product exists that could meet the barrier, mechanical, and sealing performance criteria together with cost effectiveness [12,37,38]. As the biobased packaging product industry develops, the research regarding the sustainability of biobased packaging products is developing, too. So far, there is no real consensus on how to analyze and assess their sustainability [10].

Life cycle management (LCM) is an integrated approach to consider the environmental, economic, technological, social, and economic aspects of products or organizations [39]. LCM requires companies to take a sustainability-driven approach and helps companies to take responsibility for the product [40]. As LCM is seen as a useful tool to help companies assess their sustainability efforts, it is taken as a categorization approach during the creation of the framework. A product's life cycle is considered to have three main phases: beginning of life (BOL), middle of life (MOL), and end of life (EOL). The BOL phase mainly includes design and getting the raw material, MOL includes production, use, service, and maintenance, and EOL includes reuse or disposal [27]. Kiritsis' approach to categorization [27] was used when considering the life cycle stages of the products in the framework to help assess biobased packaging products' sustainability, as can be seen in Figure 1.

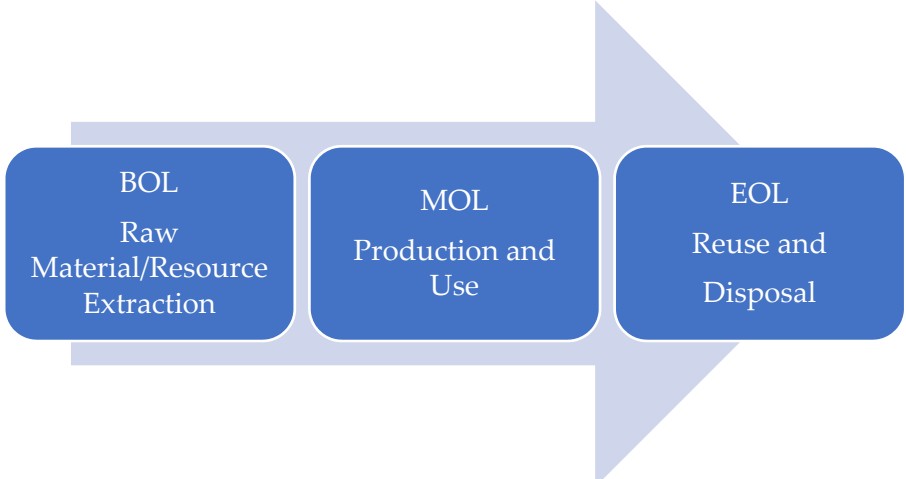

**Figure 1.** Product lifecycle of biobased plastics.

Most scholars in the area are analyzing biobased plastics in one or two stages of their life cycles by using life cycle assessment to point out the challenges [6]. There are

several scholars who have studied the overall life cycles of these products. There are also many studies focusing mainly on the end of life, i.e., biodegradation or recycling options. Especially when it comes to the social aspects, previous studies often focus on either only the problems related to usage [21], perception [20], the EOL options (consumers and society) [41,42], or the workers in the production stage [8]. Hence, it is very hard to obtain an overall picture of the social sustainability aspects.

Most of the papers in the literature review focus on the technical aspects of biobased packaging products. In the literature, the main technological challenges are defined as a lack of water vapor and gas barrier properties [19], being brittle, and having issues related to biodegradation [4]. As the focus of this study is to examine the challenges from the perspective of the Trible bottom line, technological issues will not be analyzed in detail.

The most thorough review of biobased plastics has been completed by Gerassimidou et al. [6]. This study points out the blind spots in the system and underlines the importance of feedstock selection, infrastructure availability, and the interactions between sustainability domains by making comparisons between biobased and conventional plastics. This study does not include second or third-generation feedstocks.

When it comes to researchers focusing on more than one stage of the life cycle or more than one sustainability aspect, there are a very limited number of studies. Spierling et al. [10] conducted a research review based on life cycle assessment (LCA), social life cycle assessment, and life cycle costing studies in order to fill the gap in quantifiable research on a holistic view that includes the socio-economic side of biobased plastics [10]. Even though this study analyzed several sustainability aspects of the products, it only focused on numerical life cycle assessments, excluding other assessment methods. Moreover, this study focused on cradle-to-gate studies, excluding the use and EOL phases. The study was conducted with the purpose of providing suggestions for regulatory bodies, not companies. Moreover, the study was conducted as a comparison between first-generation biobased products and conventional plastics. In the literature, most of the studies conducted are on first-generation feedstocks [6], while second and third-generation feedstocks [29,43] are often excluded. Moreover, many of the LCA studies are conducted with a cradle-to-gate approach, and the EOL section is excluded [31].

Álvarez-Chávez et al. [8] conducted research that qualitatively analyzed and compared the sustainability of different types of biobased plastics. They tried to analyze the products in terms of the safety and health of consumers, workers, and the environment. This study focused on providing suggestions for companies to make a calculated decision on which biobased products they should use. This study primarily focused on environmental health and safety, while less attention was paid to the economic and social aspects.

This study is conducted to fill the gap of creating a holistic framework to understand and manage biobased plastic packaging products' environmental, social and economic risks through the whole lifecycle, intended mainly for the companies to mitigate and manage their risks. The conceptual framework of challenges of biobased (plastic) packaging products is presented in the result section of this paper.

## 3. Methods and Descriptive Results

This study examines the overall environmental, social, and economic risks associated with biobased plastic packaging in their whole life cycle. The study explores the prior literature, comprising articles from the Scopus and Web of Science databases on bioplastics and biobased plastic packaging. These databases were selected as they represent the commonly used digital, cross-disciplinary archives for high-quality academic research [44].

To meet the study's objective, the set search words ((bio* plastic) AND packaging) AND (sustainab* OR environment* OR social OR econom*) AND (challenge OR risk OR drawback OR problem) were used.

The first search round was conducted between November 2021, and the search was complemented in March 2022. The primary data include journal articles published in academic journals, book chapters, and conference proceeding papers in order to maintain a

thorough coverage of the literature written in English. The database search was based on abstracts, keywords, and article titles. The time limit set was 2011–2021 in order to cover recent studies in the area. In March 2022, it was observed that several other articles were published within 2022, and, as a result, we decided to include these articles in the analyses, too. The summary of the research protocol is shown in Table 1.

**Table 1.** Summary of the research protocol.

| Research Protocol | Description |
|---|---|
| Databases | Scopus, Web of Science, |
| Search field | Title-Abstract-Keywords |
| Search string | ((bio* plastic) AND packaging) AND (sustainab* OR environment* OR social OR econom*) AND (challenge OR risk OR drawback OR problem) |
| Language | English |
| Data range | Beginning of 2011 until April 2022 |
| Publication type | Peer-reviewed journals, book chapters, and conference papers |
| Inclusion criteria | Papers that mention environmental, social, and/or economic challenges of biobased plastic packaging products |
| Exclusion criteria | Papers that do not address any sustainability challenges in the abstract, keyword, or title, or papers that do not contain biobased plastic packaging products |

The research process is summarized in Figure 2. The Web of Science returned 156 results, and Scopus returned 188 results. After duplicates were removed, a total of 232 documents were chosen for the abstract and title check. The abstracts and titles were checked with economic, environmental, and social sustainability in mind. After checking the abstracts and titles, 32 papers were chosen for detailed reading. After the eligibility was analyzed, 29 papers were chosen for the final check. Furthermore, these articles were subsequently checked for references in order to search for relevant important articles that may have been missed out in the search. As a result, five additional articles were added to the study in accordance with the backward snowballing strategy. Hence, a total of 34 articles were chosen for the analysis.

As can be seen from Figure 3, the number of articles published on the topic has been increasing significantly since 2017. In this research, articles published in 2022 only consider the first three months of the year. Hence, it can be expected that the number of articles published in 2022 will exceed the number of articles published in 2021 by fitting in the trend of increasing published articles. This increase in the published articles proves the increasing interest in the topic in academia and that this article is relevant and needed for further review of the studies to critically analyze the state of art in this field.

The data were analyzed qualitatively, utilizing content analysis, which is considered to provide a flexible method for analyzing textual data [45]. This process included the phases of data reduction, data display, the drawing of conclusions, and verification [46]. The analysis followed directed content analysis (see, e.g., [47]), in which the division into different life cycle stages (from BOL to MOL to EOL) was used as the starting point for the analysis of the data. Furthermore, the framework of the triple bottom line (economic, environmental, and social perspectives) guided the first level of coding.

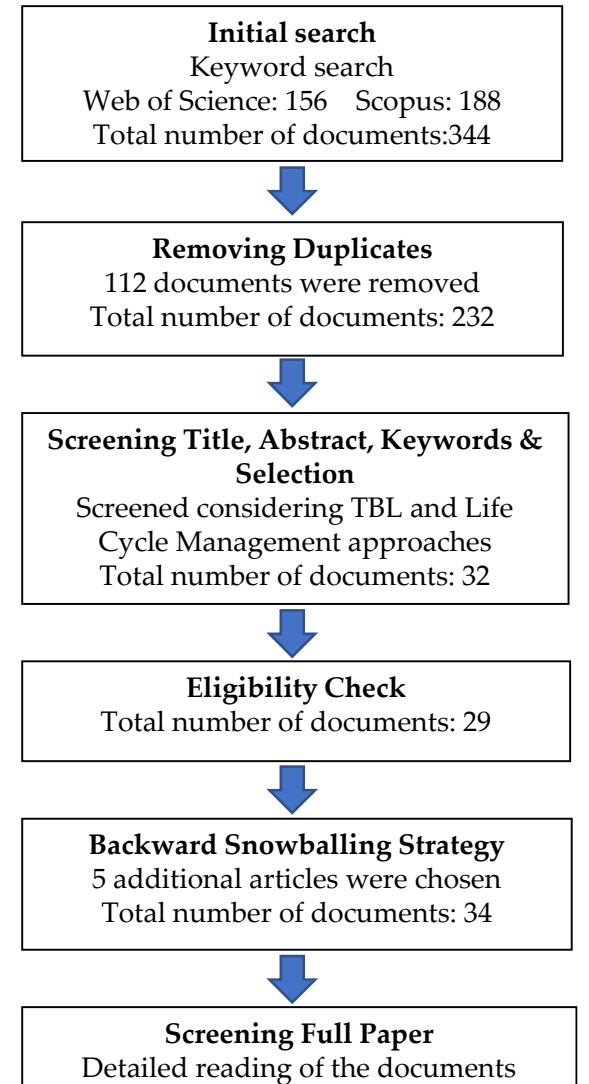

**Figure 2.** Research process.

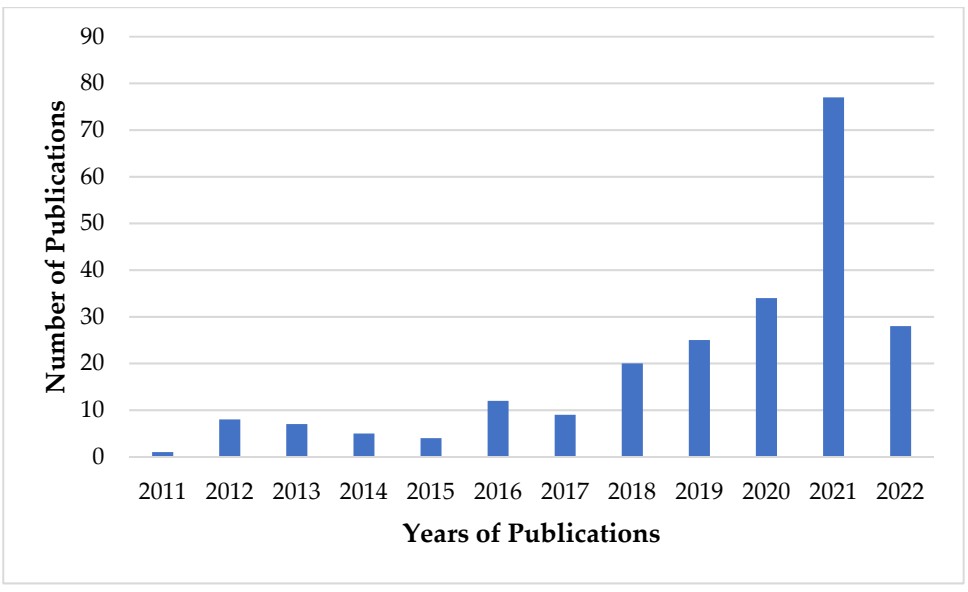

**Figure 3.** Number of publications distribution by year.

## 4. A Conceptual Framework for the Challenges Associated with Biobased Plastic Packaging

Overall, the literature on the topic focuses mostly on the technical challenges, opportunities, or possibilities related to technical product development. The review of the prior literature showed that research focusing on environmental, economic, and social challenges is limited. There are several studies focusing on the environmental challenges by conducting LCA studies on one or two life cycle stages. However, LCA studies that focus on the whole life cycle of this subject are very limited. Research on economic and social challenges is almost non-existent. In Table 2, the conceptual framework of challenges of biobased (plastic) packaging products, the identified challenges are summarized based on the life cycle phases they are connected: BOL, MOL (production and use), and EOL. Furthermore, the framework also includes some general challenges associated with several life cycle stages. The conceptual framework was formed on the basis of the literature analysis and also includes the categorization of the challenges in relation to the different sustainability aspects (economic, environmental, and social). It must, however, be noted that the categorization is not exact; different challenges may impact multiple categories.

**Table 2.** A conceptual framework of challenges of biobased (plastic) packaging products.

| | Environmental Challenges | Economic Challenges | Social Challenges |
|---|---|---|---|
| Phase 1: Beginning of life: Resource Extraction | • Land use change decreasing space for food [6,8,10–19] <br> • Loss of biodiversity [6,15] <br> • Toxic pesticide and fertilizer use can create pollution [6,8,11,13,15–17] <br> • Soil erosion and deforestation [6,15] <br> • Eutrophication [6,12,15] <br> • Use of GMOs [6,8,17,19] <br> • Potential hazard in the workplace [8,17] <br> • Decreasing water security [6,13,17] <br> • Land use change increasing carbon emissions [6] | • Possible increase in food prices with the land use change [6,16,17] | • Toxic pesticides causing risk to human health [8,15,16,19] <br> • Risk of exploitative working conditions in developing countries as they have low social standards [6,10] <br> • Alienation and negative impacts on livability [10] <br> • Land use change can alter ecosystems, increasing risk of zoonotic and infectious diseases [6] |
| Phase 2: Middle of Life: Production | • Efficiency in the use of water and energy [6,8,17,19] | • Because of the small market, redesign of production not being viable [48] | • Occupational health and safety hazard problems [6,8,17] |
| Phase 3: Middle of Life: Use | • Use of hazardous chemicals or petroleum-based co-polymers during production and processing [8,17,19] | • Low willingness to pay [21] <br> • Low demand [12] | • Low willingness to pay [21] <br> • Low demand [12] <br> • Consumers preferring other types of packaging [48] <br> • Added chemical substances during production can migrate from packaging to food and can cause human health problems [6] |

**Table 2.** *Cont.*

| | Environmental Challenges | Economic Challenges | Social Challenges |
|---|---|---|---|
| Phase 4: End of Life | • PET streams being contaminated with PLA causing poor-quality recycled PET [12,15,19,48]<br>• Non-existing standards about biodegradability in water [17]<br>• Lack of sorting infrastructure [21]<br>• Lack of biodegradation/composting/recycling infrastructure [17,37,49,50]<br>• Some packaging not degrading in marine environment for a long time [19]<br>• Degradation of biodegradable plastic increasing GHG emissions when disposed into landfills [11,15,16]<br>• Biodegradable materials contaminating organic solid waste treatment [51]<br>• Biodegradation of certain plastics can release toxic substances that can pollute environment [6,19] | • Because of the small market, redesign of waste management system not being viable [6] | • Differences in waste management systems in different areas and countries [20]<br>• Consumers not knowing how to separate waste [6,17,20,21]<br>• Labeling not being clear and reliable [49,51]<br>• Compostability claims as a license to litter [6,19,50]<br>• Biodegredation of certain plastics can release toxic substances that can create risks to human health [6] |
| General | • Certification/label not being clear [6,19,51]<br>• Not being necessarily sustainable [13,16,19]<br>• No viable option that can meet performance criteria and is cost effective [12,17,37,38]<br>• Circular economy debate on biodegradability [52] | | |

## 4.1. Economic Challenges

In the literature, there are not many studies discussing the economic challenges related to biobased products. In this paper, all the possible studies are examined within the scope in order to cover possible economic challenges mentioned in the literature. The economic problems associated with biobased plastic packaging products are linked to being more expensive than conventional plastics in general. PLA, starch, and PHAs are seen as the most economically viable materials for bioplastics, as they can be processed with pre-existing conventional converting equipment. There is also a demand for them, as they have satisfactory functional properties [53]. Because the market volume is small, major investment or the redesign of production and waste management systems is not economically viable [31]. However, if the market volume increases as expected, investment and redesign costs are expected to increase over time, making changes in the waste management system feasible.

The literature discusses how production costs can be reduced by economies of scale. However, because supply chains are long, increasing production volumes is likely to be challenging [54]. Moreover, a low production yield makes it hard to decrease prices [12]. Hence, even though a decrease in their price is expected, it remains unclear as to whether they will be priced competitively in relation to conventional plastics. Another discussion on the economic side concerns whether biobased products have the potential to affect food prices. The space they take up for planting may compete with food production space. The potential for less space for food production may create fluctuations in food pricing [1,10]. Overall, biobased polymers are more expensive than conventional solutions because the biorefinery (processing) part of the supply chain and feedstock acquisition is rather expensive [6]. High prices of biobased plastics create an economic barrier in the market, making it harder for them to become more widespread [11].

### 4.2. Environmental Challenges

There are environmental problems related to causing ecotoxicity and human health issues related to the use of hazardous chemicals or petroleum-based co-polymers during production and processing in order to increase brittleness, high oxygen permeation, and poor thermal properties [19], as well as a lack of efficiency in the use of water and energy during biomass production and processing [8,11]. Moreover, from the LCA perspective, it is not always clear what type of EOL option is environmentally better [1]. Nevertheless, the choice of EOL scenario has an important effect on overall environmental sustainability [4].

In the BOL stage, one important environmental observation concerns the need for feedstock and farming practices and their negative effects. The sustainability of a biobased product depends on the feedstock type used in the production. If the crops are first generation or second–third generation (non-edible), byproducts, residues, or wastes make a difference in terms of sustainability [15]. If the biomass used is from first-generation stocks, crop cultivation conditions become important factors; for example, the land used for the crop type and the amount and type of pesticides and fertilizers used [13]. There are also problems with biobased plastics that stem from the biobased feedstock growth creating issues due to the toxic pesticides used, which can pollute water and soil and have an impact on wildlife [8]. When it comes to eutrophication and stratospheric ozone depletion, biobased products perform worse than their petroleum-based plastic counterparts [15]. The use of GMOs to grow feedstock is a problem, as the effects of GMOs are not well understood and can lead to human health issues [8]. Water security may also be a problem in certain areas, such as Brazil and Asia, where most of the biomass is grown for biobased plastic production [6].

If the crops used are the first generation, the space used for planting may compete with food production space. Land use change can cause problems such as biodiversity loss, soil erosion, eutrophication, and ecosystem change [6]. Land use change also has the potential to increase carbon emissions. If the global biobased plastic consumption increases by 5%, carbon emissions related to the land use change could be offset in 22 years [55]. When many scholars assume that biobased products are carbon neutral, as the growing biomass removes carbon from the atmosphere, they often do not calculate land use change [10]. If the demand for biomass increases, there will be land use change causing deforestation, as new land may need to be opened by eliminating existing forest space [15]. The use of wood as a feedstock has brought with it issues in terms of deforestation [19].

In the MOL stage, efficiency in the use of water and energy in production is one of the crucial problems [6,8,17,19]. The biorefinery and polymerization processes that are used for converting feedstock to plastic are known to be highly energy intensive, sometimes more energy intensive than conventional plastic production [6], and the production of certain biopolymers such as PLA and TPS (Thermoplastic Starch) creates a high amount of wastewater [56] The chemicals added during production that can be toxic and environmentally harmful to wildlife when the products are discarded in the environment [17].

Some of the most pressing issues belong to the EOL stage. From the LCA perspective, according to some researchers, it is not clear what type of EOL option is better [1]. However, the choice of EOL scenario has an important effect on the overall environmental sustainability [4]. The plastic recycling system is already complicated and does not work well enough. An increase in biobased plastic packaging can increase the complications in terms of the identification and sorting process. Strong, clear labeling is needed [56].

The main issues related to the EOL options of these products often concern the regulations, lack of infrastructure, or customer use. This situation is also an issue in terms of checking the sustainability of the products, as it is often unknown whether the intended end-of-life design works in the real world [4]. In general, because the market share of bioplastics is still relatively small, large-scale sorting and recycling infrastructure often does not exist. Moreover, different countries have different infrastructures or lack infrastructure, especially when it comes to biodegradable plastics. For example, in Germany, biodegradable plastics are often incinerated, and composting is uncommon because biobased plastics

are recognized as contaminants in the sorting facilities [4]. Therefore, not knowing the countries in which the products will be used makes it even more challenging for companies to estimate the EOL risks.

Even though many products on the market are labeled as biodegradable, the exact composition of the product is not explained. The contents of the polymer and the additives used in the product determine the biodegradability and the time needed for biodegradation. Some biodegradable products can only degrade in certain composting facilities and not in the biomass section of the municipal waste. This can create a problem, as some biobased products can contaminate municipal waste [51]. For the municipal composting facilities to address compostable packaging and not only home biowaste, certain technical modifications will be required, particularly at the level of pre-processing, to ensure an efficient composting process for the packaging [57]. Additionally, if biodegradable products end up in landfill and do not degrade in facilities, they can increase greenhouse gas emissions. The degradation of biodegradable plastics may lead to significant greenhouse gas emissions ($CO_2$ and $CH_4$) if they are disposed of in landfills [11,56]. The dissolving of certain biodegradable materials in marine environments is also another problem. PLA can stay in a marine environment for up to one thousand years [58].

There is also a wide-reaching discussion about biodegradable plastics, according to which recycling should be the preferred EOL option instead of composting and biodegradation [15,56,59] as it creates more need for virgin materials. Moreover, there are questions raised if biobased biodegradable products follow circular economy guidelines or not [4]. The Ellen MacArthur Foundation suggests that biodegradable/compostable materials should be used only for specific applications, such as garbage bags for organic waste, and efficiently recyclable biobased products should be preferred for other uses [59].

Recycling can be a preferred alternative in comparison to biodegradation when the carbon cycle is taken into consideration. If biodegradable materials such as PHAs and PLAs could be effectively recycled in the current system, feedstock use could be decreased, a sustainable circular economy could be achieved, and the prices would drop [60]. Mendes and Pedersen, Briassoulis et al., and Hottle et al. [15,48,56] suggest that PLA, the most used biodegradable plastic packaging, can also be mechanically recycled. Additionally, according to an LCA conducted, PLA should be the preferred option [15]. However, PLA should be separately recycled from other conventional plastics. The recycling stream for PLA does not exist.

There are also issues regarding the reliability of labeling [61], especially in relation to biodegradability, due to problems such as limited methodology, unrealistic testing conditions, a lack of guidance for employing different test materials, and insufficient statistical power [19]. For example, there are European standards for assessing the biodegradability and compostability of biobased plastics in industrial composting institutions and soil. Additionally, there is no European Standard for assessing biodegradability in water, as the variability of water conditions makes standardization difficult [32]. Moreover, biobased plastics are often labeled as "other plastic", resin identification number 7, which often makes customers think that they are non-recyclable or non-compostable [6].

The literature examines customers not knowing how to dispose of biobased products, a situation that creates a big problem. Customers often believe all bioplastics are home compostable when they can, in fact, only be recycled or composted in facilities. Many biodegradable polymers compose very slowly in home composting, and they need to be composted in industrial facilities instead [4,62]. This is especially a pressing issue for PLA, the most used biobased material for biodegradable plastics, which takes over three decades to degrade in soil [63] and needs composting in industrial conditions [4]. For this to be achieved, the quantity and sorting efficiency needs to be increased [48]. Tucker and Johnson [64] have also discussed the fact that the biodegradation of biobased plastics in nature can cause cytotoxic or phototoxic material release, both of which are dangerous for human health and the environment.

Mendes and Pedersen [15] noted that, compared to landfilling, mechanical recycling, anaerobic digestion and incineration, and industrial composting may have higher negative environmental impacts. Additionally, composting PLA that has very low macronutrients cannot create fertilizers through composting.

Some biobased biodegradable products (especially PLA) can look like conventional products (especially PET), and customers can end up sorting them in the plastic recycling bins alongside non-biodegradable conventional products. Contamination of the recycled plastic streams results in poor-quality recycled conventional plastic [19,60]. It has been observed that even a very low level of contamination in the PET waste streams can make the stream unsuitable for mechanical recycling. This also creates economic problems. There is, so far, no system available for the separation of PLA from PET. Even though it is doable, it does not exist due to the costs [65]. Moreover, some scholars have raised concerns about customers believing that compostable biobased plastics can compost in nature without the need for specific composting facilities. This behavior can increase littering and the disposal of products in nature instead of using trash cans or specific composting facilities. This phenomenon is referred to as 'the license to litter' in the literature [6,19,50].

### 4.3. Social Challenges

Social sustainability issues are usually related to the fact that feedstock is cultivated in Asia or South America and the BOL stage of the products. Because most biobased materials are cultivated in countries with low social standards and weak legislation, there are issues related to poverty and several social risks [10]. Exploitative working conditions, alienation, and the negative impacts on livability and communities are identified as social risks, especially for indigenous groups in the study conducted in Indonesia by [66]. Hence, the country of origin of the feedstock and the working conditions in those countries are important. Considering the forecasted ongoing growth in the Asian markets [10], these problems are likely to increase.

Raw material extraction and the production of biobased materials can cause risks to human health. The use of GMOs for the growth of biomass can create allergic reactions and alterations in the metabolic pathway [8]. The use of a hazardous mix of substances during production and processing can create risks to human health when the workers are exposed to these substances, as they can cause burns and irritation to the eyes, skin, and upper respiratory systems with dermal contact. They are also combustible, creating a risk of explosions in the workplace. The physical extraction of PHA is reported to expose workers to certain carcinogenic substances, which can create serious health issues, including unconsciousness and coma [6].

Labels exist to make it easier for customers to navigate the EOL options. However, labeling is often unclear, and packaging can be labeled as 'compostable', 'biodegradable', or 'biobased'. The meaning of these terms is often confusing for customers [49]. Hence, even though the products are labeled correctly, bad practices can exist due to confusing labeling [32]. Moreover, many products in the market are labeled as biodegradable, but the composition of the polymers and additives are not provided. This composition influences biodegradability and the time needed for degradation. As a result, consumers do not know how to dispose of the product [50].

According to a study conducted in Australia by Dilkes-Hoffman et al. in 2019 [41], public knowledge of biobased plastics is so low that some customers assume biobased plastics are always biodegradable. However, the perception is positive as they are perceived as being better than conventional or recycled plastics. In a study conducted in Germany by Klein et al. [41] reached the conclusion that biobased plastics are perceived well by customers. Herbes et al. [67] conclude in their study that biodegradability is preferred to only being biobased. The research also shows that customers are not aware of how to dispose of biobased plastic packaging. In the study conducted by Dilkes-Hoffman et al. in 2019 [41], 62% of the customers said that they would dispose of biobased plastics in a regular recycling bin. In a study conducted by Taufik et al. [20], being biobased and

biodegradable confused customers as they believed plastics, even though biobased, can be only recyclable. Customers' high willingness to buy biobased packaging products even though they do not know how to properly dispose of them shows a clear challenge about the EOL of these products.

One important aspect regarding biobased packaging is that instead of biobased, some consumers may prefer recycled plastic or paper/glass packaging over biobased plastic packaging [49].

## 5. Discussion and Concluding Remarks

The literature clearly shows that the sustainability of biobased plastic packaging depends on many factors, such as the type of feedstock used, the farming conditions during the growth of the biomass, the available EOL options, and if the end users can appropriately dispose of these products.

The first attempt in this study was to create a literature review that focuses on all biobased packaging types. However, when the search words "biobased packaging" or "bio-based packaging" were used, almost all the answers were related to "biobased plastic packaging". This shows that most research about biobased products is on biobased plastics and their challenges. Even though wood or bamboo fiber packaging and even some forms of paper-based packaging are considered to be biobased, they were rarely represented in these specific keyword search results. Hence, keywords are revised to be "bio* plastic packaging" in order to cover the literature on the biobased plastic packaging area.

Different domains of sustainability challenges can interact with each other. Environmental problems in resource extraction, such as decreasing water security, soil degradation, and eutrophication, can have a negative impact on human health, and this can cause economic challenges due to decreasing human labor power. The economic challenge of a possible increase in food prices can lead to malnutrition and have effects on local communities (social). A low willingness to pay (social) during the use phase can prevent the product prices from dropping further, and this keeps the market small. As the market is small, redesigning production would not be viable (economic), and the infrastructure cannot be developed to have better recycling systems (environmental). Social challenges in the EOL phase, such as consumers not knowing how to separate, would create environmental challenges by waste streams being mixed and recycled materials being of poor quality.

The main identified challenges of biobased packaging products were related to biomass cultivation and farming that can create water stress, pesticides creating environmental pollution and health risks, possible land use change causing competition with food production and thus causing problems with food availability and high food prices, biodiversity and deforestation, customers not knowing how to dispose of products, and the lack of infrastructure for end-of-life options. To manage these challenges, better regulation and control at all stages of the life cycle, companies taking responsibility to follow their supply chain and choose responsible business practices, as well as communicating the correct disposal options to their customers, better labeling and education of customers are needed.

In terms of the theoretical implications, this study has increased our comprehensive understanding of the challenges associated with biobased packaging products. The study not only highlights the different challenges in different life cycle phases but also connects them with different sustainability dimensions going beyond the state of the art of the research in the field (e.g., Gerassimidou et al. [6]), giving recommendations for companies and pointing out the challenges associated with biobased plastic packaging products without comparison to other products. This provides a starting point for the development of different assessment tools to assess the sustainability impacts of different packaging products. Based on this, this study raises important questions that provide avenues for future research on more detailed data, as well as assessment tools to assess the sustainability of biobased packaging products regarding the different sustainability aspects.

This study is one of the first to address the social perspective on the challenges associated with biobased packaging products. Based on these, we can ask, e.g., the following:

- What are the working conditions in the raw material extraction/ production stage?
- How is the social community affected by biomass growth?
- What can be done to increase consumers' awareness of the correct EOL options?

Expanding the research to cover qualitative data from consumers and companies would provide a fruitful avenue to examine the challenges regarding production, usage, and EOL habits. Moreover, expanding the research about what the mitigation strategies are for the mentioned challenges and what and how much actually the companies are doing to mitigate the mentioned challenges would provide a fruitful avenue for future research.

The present study concluded that the economic challenges associated with the products in question are not well presented in the literature, besides brief mentions that they are expensive. What can be done to decrease their prices? Our study also concluded that there are not many articles addressing the use phase of these products. It would be important to conduct further research into the possible environmental and health effects of the use phase

With regards to managerial implications, the conceptual framework presented in the study provides a tool to identify the possible challenges associated with the products that companies are developing. Thus, this is a framework to anticipate possible challenges and risks and to develop additional tools to manage them. This tool can be used by company operators to manage biobased packaging challenges. Moreover, being aware of these challenges can help them more effectively communicate the specifics of biobased plastics to their customers. Communicating biobased plastics as if they are truly sustainable without being aware of their challenges can create the risk of greenwashing customers and reputational risks. Moreover, knowing that there are social risks related to the customers, that there are issues regarding labeling not being clear, and regarding customers not knowing how to sort the waste correctly can help businesses design packaging for the easy and correct use/disposal of their products. Furthermore, being aware of the issues related to the lack of infrastructure and regulations changing in different countries can help businesses take these issues into consideration during the design process.

Design that takes into account the EOL challenges is needed. Customers often do not know how to separate waste correctly, or they do not know how to read the labels. Some countries do not even have the necessary infrastructure. Considering where the products will be sold, recycling and composting infrastructure in the targeted sales country should be considered while designing the product. If there are no composting or biodegradation facilities, or if the customers cannot easily reach them, recyclable products in the current system should be preferable. When recyclable products are designed, all efforts should be made to ensure that the type of product can be recycled in that country's system. Moreover, making sure labels and disposal instructions are clear and easy for the customers to understand and follow is an important aspect for the companies.

In conclusion, biobased packaging products are not always more sustainable. Companies need to analyze their product life cycles carefully, be aware of the possible challenges and act accordingly.

As is the case with all studies, this study also has its limitations. This study is limited by the fact that it does not cover empirical data; instead, the analysis was based on peer-reviewed articles complemented with selected gray literature supporting the knowledge gaps. Furthermore, there may also be additional relevant articles in other types of literature sources, such as government reports or project evaluation reports, that would provide insights for the analysis. Although the research was complemented by Google searches for gray literature, there was a high degree of variety in the documents (such as commercial reports and news articles), and this increased the number of irrelevant articles. Thus, the detailed analysis of all the search results was considered unnecessary. Furthermore, the set search words have their limitations. The review revealed that the words "bioplastic packaging" or "biobased plastic packaging" are used almost synonymously with "biobased packaging". Hence, there may be certain articles that work on biobased plastic packaging by using the term "biobased packaging". Those articles could not be included in the data set, as the search words were "bio* plastic packaging". Moreover, there may be articles that work

on 'biobased products' without mentioning 'packaging', and these are excluded from the data set. To limit the search results with the aim of focusing on packaging specifically, search words are limited. In this expanding research field, further detailed systematic reviews of literature with possible citation/co-citation analyses could also provide a potential avenue for future study.

**Author Contributions:** Conceptualization, D.T.; data curation, D.T.; formal analysis, D.T.; funding acquisition, N.T. and V.O.; investigation, D.T.; methodology, D.T.; resources, D.T.; supervision, N.T. and V.O.; validation, D.T.; visualization, D.T. and N.T.; writing—original draft, D.T.; writing—review and editing, D.T., N.T., and V.O. All authors have read and agreed to the published version of the manuscript.

**Funding:** This research was funded by NordForsk as a part of the research project Citizens as Pilots of Smart Cities.

**Institutional Review Board Statement:** Not applicable.

**Informed Consent Statement:** Not applicable.

**Data Availability Statement:** Not applicable.

**Conflicts of Interest:** The authors declare no conflict of interest.

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
