# Peer review of "A Conceptual Framework of the Sustainability Challenges Experienced during the Life Cycle of Biobased Packaging Products"

_sustainability, doi:10.3390/su141710465_

Round 1

Reviewer 1 Report

The authors are encouraged to continue to address this extremely interesting and necessary topic for all the mankind.

Author Response

Dear Reviewer,

Thank you very much for your comments.

Reviewer 2 Report

The manuscript entitled "A conceptual framework of the sustainability challenges experienced during the life cycle of biobased packaging products" proposes an interesting subject to ensuring a sustainable ecosystem to avoid critical economical and health issues. The main purpose and approach generally coincided with the scope of the journal. Overall, the main idea of the manuscript is interesting. However, I have comments that authors should address, listed below:

1. From the introduction, I’m not sure about the framework of the review and how it is constituted to the relevance of the review. Perhaps authors should design a well constructive diagram, detailing the framework of how biobased packaging products seek to address the 3 main challenges (environment, social and economic). It will be easy to understand and captivating for the reader's clarity. The first figure should be that of the framework.

2. Section 2.1, Line 139. Not all microorganisms have biodegrading functions, hence authors must be somehow specific and list those involve in high degrading activities. “microorganisms such as (…..)”.

3. Figure 1 is not clear and must be improved. It should be reproduced with quality resolution and size since it’s difficult to read.

4. Section 2.2, Line 168. Change the punctuation after the reference [6,10] before “while”. It should be a comma

5. Section 2.2, Line 171-173. The paragraph is too short and should be added to the previous paragraph.  

6. Talking about the life cycle in Section 2.2, the authors can produce a figure for this section to highlight the life cycle management processes. This will further capture the reader's attention and enlighten their understanding of the subject.

7. The experimental design is very critical to the output of the review, hence figure 2 must be redrawn at the highest resolution. It is difficult to read and blurs out when zoomed out and the same for figure 3.

8. How was the eligibility check in figure 2 conducted? The authors should briefly explain the backward snowballing strategy.

9. Section 3, Line 275. If the words BOL, MOL and EOL are initials then they should be defined in the first mention.

10. Section 4.1. How do the authors conclude the paragraph if there are not many related documents? The paragraph requires a conclusion. Same for the second paragraph.

11. Section 4.2, Line 320. Remove one “use of”’

12. Section 4.2, Line 356. PLA and TPS including all other initials used in the manuscript should be defined on the first mention. Line 387. CO2 and CH4, the numbers should be subscripted

13. Some aspects of the manuscript such as section 4 lack possible solutions to the challenges raised. As a review manuscript, authors should provide the possible solutions to questions raised other than just quoting other references and research.

General comments: The review was well written, highlighting the core of various categories related to biobased products. However, there are too many short paragraphs in the manuscript such as in section 2.2 and most paragraphs lack conclusions. Also, the authors should improve the resolutions of all the figures in the manuscript and those that can be reproduced should be made accordingly.

Author Response

Dear Reviewer,

You may find the response to your comments in the attached document.

Mentioned changes in the document are done with red ink in the updated manuscript.

Reviewer 3 Report

Many papers and many reviews revolve around biobased plastics. I had the same feeling that most of them revolves around technical problems and that less papers reports on a global view including not only technical issues akin to environment but also the economical and the social aspects. It is interesting that this feeling is confirmed by a systematic bibliography study. Such a study fits perfectly with the field of “sustainability”.

Accordingly, it raises important questions for the development of a sustainable policy for biobased polymers. I completely validate the points discussed in the paper. 

COMMENTS

In this field, there is a lot of confusion on definitions, and I think it is useful to clarify these definitions.

1)    A lot of confusion was present in the past on the definition for biodegradation. For this topic, the definition has been clearly detailed in the paper and I fully validate the definition used by the authors for biodegradation.

2)    If I correctly read the paper, the authors considers that biopolymers are bio-sourced polymers and not bio-degradable polymers. It was not always so obvious in the past because some authors considered bio-polymers as biodegradable polymers. I completely understand the authors because the current trend is to limit biopolymers to bio-sourced polymers. Nevertheless, I am not aware of an official definition used by all the scientific community. The paper can be updated in 2 different ways. (1) the authors explains the possible confusion and clearly indicate the definition they use the first time they mention bio-polymers in their paper (2) they avoit ot use the word bip-polymer and they stick to bio-sourced polymers and bio-degradable polymers.

3)    The concept of recyclability could be better defined because it can include the physical recycling (without modification of the chemical structure of the polymers) or chemical recycling (or chemical valorization, thus with chemical modification of the chemical structure of the polymers). The definition can be a little vague because, potentially all polymers are recyclable in theory but not all of them can be efficiently recycled  in the real life by considering the technical, environmental, economic, and social aspects. My feeling is that this point is important because many companies could claim their products recyclable (or even 100% recyclable) even though they are not recycled or only partially recycled.

4)    For bio-sourced polymer, it could be clearly defined that a bio-sourced polymer is a polymer synthesized from biomass as the harvested source of carbon in nature. By using this definition, oil is not included in the biosourced polymer even though it is produced in nature from biomass. In other words, the definition does not take into account any source of the carbon because there is no source in the cycle of carbon present in nature as everybody knows.

 Line 46 - The sentence “Main of these disadvantages stem from the increasing scarcity of fossil fuels” is not very clear” because the reader could understand the oil is already scarce today, which is not so obvious to-day. Of course, the sentence is right because oil resources are not renewed at the same speed oil is harvested by human beings, and, even though, this in sot the case right now, oil will become scarce in the future. An economic point of view could claim that oil is not so scarce because the price of oil-sourced products remains competitive with bio-sourced polymers. A slight change of the sentence could avoid the confusion.

Line 134 – polyethylene and polyethylene, polypropylene can be recycled (physical recycling) and the authors wrote the opposite. Maybe, the authors are dealing with chemical recycling. This confusion should easily be raised by adjusting the definitions.

Figure 1 - I agree with the data shown by the figure, which is very classical in the field. Nevertheless, a short notice could indicate that the graph could evolve with time. For instance, patents are reported for the production of bio-based PCL even though PCL remains produced from oil at the time being in industry as shown in Figure 1.

Author Response

Dear Reviewer,

Thank you very much for your valuable comments. You may find our response in the document attached.

We would like to mention that referred changes are made with the blue ink in the updated manuscript.

Round 2

Reviewer 2 Report

The revised manuscript is okay

Author Response

Dear Reviewer,

Thank you very much for your your comment.